# Removal of an Azo Dye from Wastewater through the Use of Two Technologies: Magnetic Cyclodextrin Polymers and Pulsed Light

**DOI:** 10.3390/ijms23158406

**Published:** 2022-07-29

**Authors:** María Isabel Rodríguez-López, José Antonio Pellicer, Teresa Gómez-Morte, David Auñón, Vicente M. Gómez-López, María José Yáñez-Gascón, Ángel Gil-Izquierdo, José Pedro Cerón-Carrasco, Grégorio Crini, Estrella Núñez-Delicado, José Antonio Gabaldón

**Affiliations:** 1Molecular Recognition and Encapsulation Research Group (REM), Health Sciences Department, Universidad Católica de Murcia (UCAM), Campus de los Jerónimos 135, E-30107 Guadalupe, Spain; mirodriguez@ucam.edu (M.I.R.-L.); japellicer@ucam.edu (J.A.P.); tgomez@ucam.edu (T.G.-M.); daunon@ucam.edu (D.A.); vmgomez@ucam.edu (V.M.G.-L.); mjyanez@ucam.edu (M.J.Y.-G.); jceron@ucam.edu (J.P.C.-C.); enunez@ucam.edu (E.N.-D.); 2Research Group on Quality, Safety and Bioactivity of Plant Foods, Department of Food Science and Technology, CEBAS-CSIC, University Campus of Espinardo, Edif. 25, E-30100 Espinardo, Spain; angelgil@cebas.csic.es; 3Laboratoire Chrono-Environnement, UMR 6249, UFR Sciences et Techniques, Université Bourgogne Franche-Comté, 16 Route de Gray, 25000 Besançon, France; gregorio.crini@univ-fcomte.fr

**Keywords:** β-cyclodextrins, porous adsorbent, adsorption kinetics, organic contaminants, advanced oxidation process

## Abstract

Water pollution by dyes is a huge environmental problem; there is a necessity to produce new decolorization methods that are effective, cost-attractive, and acceptable in industrial use. Magnetic cyclodextrin polymers offer the advantage of easy separation from the dye solution. In this work, the β-CD-EPI-magnetic (β-cyclodextrin-epichlorohydrin) polymer was synthesized, characterized, and tested for removal of the azo dye Direct Red 83:1 from water, and the fraction of non-adsorbed dye was degraded by an advanced oxidation process. The polymer was characterized in terms of the particle size distribution and surface morphology (FE-SEM), elemental analysis (EA), differential scanning calorimetry (DSC), thermal gravimetric analysis (TGA), infrared spectrophotometry (IR), and X-ray powder diffraction (XRD). The reported results hint that 0.5 g and pH 5.0 were the best conditions to carry out both kinetic and isotherm models. A 30 min contact time was needed to reach equilibrium with a qmax of 32.0 mg/g. The results indicated that the pseudo-second-order and intraparticle diffusion models were involved in the assembly of Direct Red 83:1 onto the magnetic adsorbent. Regarding the isotherms discussed, the Freundlich model correctly reproduced the experimental data so that adsorption was confirmed to take place onto heterogeneous surfaces. The calculation of the thermodynamic parameters further demonstrates the spontaneous character of the adsorption phenomena (ΔG° = −27,556.9 J/mol) and endothermic phenomena (ΔH° = 8757.1 J/mol) at 25 °C. Furthermore, a good reusability of the polymer was evidenced after six cycles of regeneration, with a negligible decline in the adsorption extent (10%) regarding its initial capacity. Finally, the residual dye in solution after treatment with magnetic adsorbents was degraded by using an advanced oxidation process (AOP) with pulsed light and hydrogen peroxide (343 mg/L); >90% of the dye was degraded after receiving a fluence of 118 J/cm^2^; the discoloration followed a pseudo first-order kinetics where the degradation rate was 0.0196 cm^2^/J. The newly synthesized β-CD-EPI-magnetic polymer exhibited good adsorption properties and separability from water which, when complemented with a pulsed light-AOP, may offer a good alternative to remove dyes such as Direct Red 83:1 from water. It allows for the reuse of both the polymer and the dye in the dyeing process.

## 1. Introduction

The constant development of industrial companies and the excessive employment of chemical compounds have led to environmental pollution. Due to this trend, a large amount of colored wastewater is discharged into aqueous effluents from textile industries [1,2]. Wastewaters rich in dye entities are complicated to handle since dye molecules are nonbiodegradable, recalcitrant contaminants incompatible with aerobic digestion, and resistant to oxidizing agents. In addition, dyes are persistent and might lead to mutagenic precursors including teratogenic and carcinogenic agents [3]. Their release into the environment produces serious problems in ecosystems such as a reduction in photosynthetic pathways, a reduction in oxygen levels, and even suffocation of fauna and flora [4].

The classification of dyes includes cationic, anionic, and nonionic based on the properties they impart to water solutions. Azo dyes are especially toxic because of the presence of amine groups in the structure. These azo dyes might induce damage to both the environment and human health, and their levels should be diminished as much as possible before releasing them into water effluents [5].

Numerous techniques including chemical oxidation [6], adsorption/biosorption [7,8], coagulation/precipitation/flocculation [9], oxidation processes, ion exchange, biological degradation, and membrane filtration have been proposed for dye removal from wastewater. Unfortunately, most of these approaches are not practicable in industrial frameworks because of several main drawbacks such as high operational costs, constricted environmental conditions or the production of toxic by-products. In practice, there is no single process that can provide a successful treatment, and the simultaneous use of different processes is required (e.g., coagulation-precipitation associated with an adsorption/filtration process is implemented) to economically reach the required water quality. Thus, there is a need to generate new decolorization strategies that are effective, cost-attractive, and acceptable in industrial use [10,11].

Adsorption is considered as a suitable technique for dye removal. The major pros of this technique rely on its simplicity of design, high efficiency, low cost, nontoxic by-products, fast adsorption rate and wide adaptability. Recently, many approaches have attempted to produce cheaper and effective adsorbents containing biopolymers for dye removal. Indeed, it is now well-known that adsorption methods based on biopolymers are effective and economic alternatives for cleaning water [5]. Among them, cyclodextrin-based polymers have attracted considerable attention because they have better adsorption capacities and stereoselectivities than the parent cyclodextrin because of their particular macromolecular structure and larger number of cavities in the polymeric network, providing more interaction sites. Cyclodextrins (CDs) are obtained through the enzymatic degradation of starch and are composed of an internal cavity that is hydrophobic and can contain a different number of glucose units joined together by glycosidic bonds [12,13].

The hydrophobic cavity of CDs permits the formation of inclusion adducts through host–guest interactions. Due to these properties, CDs can provide the formation of stable complexes with different compounds such as dyes. These special properties attributed to CDs can be used in many areas and industries. The ability of CDs to yield stable complexes with many wastewater pollutants has been extensively demonstrated [12,14,15].

In general, the formation of insoluble CD-based materials depends on polymerization with a cross-linking agent. However, CD polymers are almost nonporous structures and possess low adsorption rates. In this study, CDs are green extractants and were selected as compounds to be used in the preparation of magnetic polymers, and the polymerization of CDs was achieved by using epichlorohydrin (EPI). EPI is a flexible cross-linking agent that permits the production of insoluble, biodegradable, and environmentally friendly polymers [16]. Nevertheless, the problems associated with the separation of adsorbent polymers from water solution hinder their application in water treatments. Magnetic nanoparticles (Fe_3_O_4_) have been extensively adapted to environmental uses (e.g., for the elimination of pollutants and the separation and purification of analytes of bioactive ingredients) due to their unique magnetic characteristics and rapid and efficient properties in the reduction of pollutants from wastewater by implementing an exogenous magnetic field [17]. Indeed, wastewater treatment and the design of CD-based magnetic adsorbents for pollutant removal have a main advantage: magnetic materials possess the characteristics of convenient separation and high extraction capacity for pollutants (they are easily separated by an external magnetic field since they have a magnetic core), which reduce the cumbersome and time-consuming filtration, sedimentation, or centrifugation steps [18,19,20,21,22,23,24,25,26,27]. Another interesting feature is the fact that Fe_3_O_4_ particles are biocompatible and are used in biological applications such as drug delivery and molecular imaging. For these reasons, we prepared and evaluated β-CD-EPI-magnetic polymers as adsorbents for dye removal [19,20,21,22,23,24,25,26,27,28].

The reduction of dyes by adsorbent polymers is not completely efficient. While some adsorbents can retain high levels of contaminants, a problem still exists regarding the amounts of contaminants that are not retained and can eventually reach the environment. To cope with this problem, the degradation by a novel pulsed light/H_2_O_2_ process of the amounts of contaminants that are not retained by the polymers might be useful. In the pulsed light/H_2_O_2_ process, hydroxyl radicals are produced photolytically from H_2_O_2_ by means of a pulsed light system, which provides a new light source to use in advanced oxidation processes (AOPs) and a new application of pulsed light technology. Pulsed light generates very intense light that includes UV emissions, which can give rise to a process much faster than those driven by conventional UV lamps, with the additional advantage of using mercury-free lamps [29].

The present manuscript analyses the adsorption phenomena of Direct Red 83:1 dye on β-CD-EPI-magnetic polymers as an adsorbent and evaluates the potential application of this polymer. Thus, the experimentally recorded data were fitted to various kinetic and isotherm models to analyze the adsorption properties of magnetic polymers. To understand the adsorption process, the adsorbent was fully characterized, and the spontaneity of adsorption was calculated by determining the thermodynamic parameters. In this way, a novel polymer, a novel light source for AOPs, a novel use of pulsed light technology and a novel array of complementary methods was tested and characterized.

## 2. Results and Discussion

### 2.1. Characteristics of the Polymer Material

The β-CD-polymer-modified Fe_3_O_4_ beads showed a specific surface area of 0.0126 m^2^/g (smaller than that of the nonmodified Fe_3_O_4_ beads (β-CDs-EPI), 0.0963 m^2^/g), a narrow size distribution (1.439 of span value), lower than β-CD-EPI (2.256 of span value), and possessed a higher particle volume D[4,3] = 641.1 µm versus D[4,3] of 136.6 µm for β-CD-EPI, expressed as the mean volumetric size D[4,3]; the results are shown in the Appendix A. The corresponding carbon, hydrogen, nitrogen, and sulfur contents were also obtained, revealing that considering the elemental content of β-cyclodextrin (C = 45.47%; H = 7.25%), the results after polymerization showed a slight increase of 2% in carbon content due to the EPI crosslinker (C = 47.80%; H = 7.11%). However, when polymerization was carried out in the presence of Fe_3_O_4_ nanoparticles, the carbon content dropped by almost half (β-CD-EPI polymer-modified Fe_3_O_4_ nanoparticles: C = 25.26%, H = 3.88%). FE-SEM was used to characterize the morphology of iron containing the β-CD polymer. Two different samples were examined, the raw polymer (Figure 1A), and the powder reduced sample (Figure 1B) used for X-ray diffraction.

Both samples appeared with a heterogeneous surface morphology and size distribution (Figure 1A,B). No defined shapes could be drawn from the images obtained, suggesting an irregular structure for the polymer.

The corresponding EDX spectra were also taken, and are shown in the Appendix A. The coating of the epichlorohydrin-β-cyclodextrin polymer on the surfaces of the Fe_3_O_4_ particles was verified by IR spectroscopy (Figure 2B). The results showed a wide band centered at approximately 3400–3300 cm^−1^ in the iron-containing and non-containing polymer, corresponding to the stretching modes of vibration for the O–H bonds of the primary, secondary, and isopropyl nonbonding hydroxyl groups of glucopyranose and the aliphatic side chains generated during polymerization (Figure 2A, B, D). An additional band at approximately 2870 cm^−1^ appeared in both polymeric materials (Figure 2 B, D) with respect to the band at ca. 2920 cm^−1^ in the single cyclodextrin (Figure 2A), all stretching C–H modes in the CH and CH_2_ groups of the native rings and the new backbones bonded to them.

Deformation C–O–H modes were present at 1500–1200 cm^−1^, stretching C–O–C modes at 1200–850 cm^−1^ and other nonspecific C–H deformation peaks below 800 cm^−1^ completed the spectrum for cyclodextrin, epichlorhydrin polymerized cyclodextrin, and the iron oxide epichlorhydrin cyclodextrin polymer (Figure 2A,B,D). The incorporation of metallic particles in the core of the β-CD-EPI polymer was proven by the appearance of the most relevant band of the uncoated iron oxide spectrum, the octahedral Fe–O stretching vibration band at 532 cm^−1^ (Figure 2C) in the fingerprint region of the polymer at 549 cm^−1^ (Figure 2D), which was slightly offset from the original band. Taking into account the method used in the preparation, in which the iron particles were synthesized before polymerization, and the spectroscopic data of the final material, from which we could not infer distortion in the iron oxide crystal lattice, we are inclined to think that the interactions are probably hydrogen bonds and coordination between the oxygen of the polymer to iron rather than proper chemical bonds.

The thermal behavior of the samples, profiles of the *m*/*z* ratios, experimental mass loss curve, and its first derivative are shown in Figure 3. The weight loss/gain with a change in temperature (TGA profile) is shown in the Appendix A. As shown in Figure 3A, nonpolymeric cyclodextrin suffered a first weight loss at low temperatures (120 °C), corresponding to the absorbed water molecules (*m*/*z* 18). No significant weight loss was registered up to 350 °C when almost total mass loss occurred (up to 89%). Within the mass range considered, there were a multitude of unspecific losses typical of minor polyhydroxylated alicyclic compounds as well as water and carbon dioxide, with the majority of weight corresponding to fragmentations of higher mass due to the decomposition of carbohydrate molecules.

Fe_3_O_4_ nanoparticles (Figure 3B), synthesized prior to the polymerization step with less than a 5% weight loss, were recorded up to 800 °C, with the main masses associated with the loss being water (*m*/*z* 18), oxygen (*m*/*z* 32), and carbon dioxide (*m*/*z* 44), probably due to small impurities present in the sample.

Iron-containing (Figure 3D) and non-containing polymers (Figure 3C) showed completely different decomposition patterns. On one hand, the cyclodextrin polymerized with EPI in the absence of iron nanoparticles showed a decomposition process at a slightly higher temperature than cyclodextrin alone, together with another residual process at close to the same temperature of decomposition than cyclodextrin. In any case, the possibility of consisting of unbound nonpolymeric cyclodextrin was discarded by the synthetic method, since all products were washed to remove the unreacted starting material before analysis. This fact was confirmed in the DSC analysis (Figure 4 and Figure 5). On the other hand, polymerization in the presence of Fe_3_O_4_ nanoparticles resulted in two markedly different decomposition steps, one of them (41.6% weight loss) at a temperature slightly lower than the parent cyclodextrin, which implies a slight destabilization compared to the EPI-β-CD polymer, and the other one (22.8% weight loss) at a considerably higher temperature, 650 °C, with two *m*/*z* values 77 and 78, together with a *m*/*z* at 44, indicative of carbon dioxide that probably proceeded from the carbonylation of iron atoms in the first step. Despite the fact that *m*/*z* 77 could be indicative of organic chlorinated compounds, the corresponding *m*/*z* at 36 at the basal level (picograms) during the whole process was attributed to the O-18 molecules (also present in *m*/*z* 46, OCO-18), rather than hydrogen chloride due to the high oxygen content in the polymer.

Other authors have suggested that the third weight loss observed at 650 °C could be due to the phase transition from Fe_3_O_4_ to FeO [30]. Therefore, the presence of chlorine from the cross-linking in the newly coated iron nanoparticles was discarded. Regarding the DSC analysis (Figure 5), the EPI-Fe-β-CD polymer showed a glass transition between 188 and 189 °C before any decomposition took place. This transition, attributed to some kind of relaxation of the structure to a more flexible arrangement, was absent in the EPI-β-CD polymer or in any of the starting materials.

The XRD pattern of the magnetic adsorbent is shown in Figure 6. Six characteristic diffraction peaks of Fe_3_O_4_ can be found at 2 θ = 30.2°, 35.7°, 43.5°, 53.9°, 57.6°, and 63.2°.

These can be assigned to diffractions from the (2 2 0), (3 1 1), (4 0 0), (4 2 2), (5 1 1), and (4 4 0) planes of Fe_3_O_4_, respectively, indicating that the epichlorohydrin-mediated polymerization of the cyclodextrin in the presence of the preformed iron particles did not result in any distortion of the crystal lattice (see Appendix A: degree of crystallinity of 68 for the Fe_3_O_4_ nanoparticles and 70% for the β-CD-EPI polymer-modified Fe_3_O_4_ nanoparticles), suggesting a structure in which the magnetite nanocrystals are superficially coated with the in situ formed polymer, thus preventing the formation of macromolecular aggregates.

### 2.2. Adsorbent Dosage

The effect of adsorbent dosage is shown in Figure 7. The experimental conditions were fixed at 50 mg/L of Direct Red, an agitation speed of 500 rpm, and increasing concentrations of the polymer at pH 7. The results showed that increasing the amount of polymer led to a higher dye removal up to a point in which the level of removed dye was stable. In the range from 0.5 to 2 g, the performance was similar, and approximately 85% of dye was entrapped from the water solution. For this reason, 0.5 g of adsorbent was selected to perform further experiments.

### 2.3. pH Effect

The adsorption effectiveness of substances onto polymers depends on the pH. Changes in the pH value can lead to changes in the charge of both the adsorbent surface and the ionization degree of the substance [31]. These experiments were conducted using 0.5 g of the adsorbent, 50 mg/L of dye, 500 rpm, and a pH range from 3 to 11 (Figure 8).

All of the accumulated data indicated that acidic conditions were the most suitable to perform the adsorption experiments (pH 3 and 5). In contrast, the process became less favorable in alkaline media. Indeed, the recorded UV–Vis spectra suggest that the adsorption capacity sharply decreased by 30% at pH = 11 (see Appendix A) because an increase in negative charges, both in the dye and the adsorbent (cyclodextrins located on the surface of the magnetic polymer), caused a repulsive force between the two, affecting their adsorption. As pH emerged as a critical parameter in the removal procedure, we performed theoretical calculations to further rationalize the impact of protonation states on the adsorbent/dye interaction. At an early stage of the computational strategy, we delineated the protonation states at pH = 3, 5, 7, 9, and 11. According to the developed model systems, Direct Red 83:1 mainly exists as a treat-anionic form in the range of pH = 3–7, corresponding to four sulfonate (SO_3_^−^) groups. The increase in pH is concomitant with the deprotonation of the hydroxyl substituents, which in turn yields more negative species at pH = 9 (charge = −5/−6) and pH = 11 (charge = −7/−8). The global charge significantly affects the pose adopted by the dye upon encapsulation (Figure 9), a qualitative picture that was eventually completed by assessing the interaction energy (see Appendix A).

Remarkably, we observed a direct correlation between the protonation state (charge) and the predicted interaction energy. If dye is embedded into β-CD at pH values ranging from 3 to 4, our models foresee an interaction energy of ca. −40 kcal/mol. In contrast, dye is anchored with an energy of ca. −14 kcal/mol only if the pH is tuned up. Molecular models confirm that protonation states are critical for the absorption phenomena, and appear to be a valuable tool for optimizing the experimental conditions.

### 2.4. Effect of Contact Time

Figure 10 shows that the adsorption ability increased until the adsorption of dye on the polymer stopped when equilibrium was reached at each concentration.

When equilibrium is reached, the amount of adsorbed/desorbed dye inside/outside the polymer is in equilibrium. The time needed to reach this point is the equilibrium time, and the amount of dye entrapped by the polymer indicates the maximum absorption capacity [32]. The adsorption process showed a rapid rise in the preliminary stages of contact between Direct Red 83:1 and the magnetic polymer. In the case of the adsorption of the β-CD-EPI magnetic polymer, only 30 min of contact time was needed to reach equilibrium (Figure 10). When the contact time increased, the quantity of adsorbed dye remained constant. Across the whole range of concentrations analyzed, all curves were asymptotic.

### 2.5. Kinetic Analysis

The adsorption kinetic data of Direct Red 83:1 were analyzed using three kinetic models, the pseudo-first-order, pseudo-second-order, and intraparticle diffusion, to investigate the adsorption kinetics and mechanisms implicated in the adsorption process of dye onto β-CD-EPI magnetic polymers (Table 1). The best model was selected based on the results provided by the adjustment (R^2^). The linearity of the pseudo-first model (log (*q_e_-q_t_*) versus t) was graphed for 120 min of contact time (Figure 11A).

The R^2^ values for this model ranged from 0.924 and 0.989 for β-CDs-EPI-Fe. The calculated *q_e_* differed from the experimental value of *q_e_* for this kinetic model, indicating that the pseudo-first model was not adequate to describe the adsorption process; for this reason, the pseudo-second model was applied.

The plot of *t/q_t_* versus *t* yielded straight lines across the whole range of measurements (Figure 11B). In all cases, the R^2^ values were equal to 0.999. The calculated qe value was quite close to the experimental value of *q_e_*. According to these results, the pseudo-second-order model presented the best fit to the experimental data, and it can be concluded that adsorption was favored by chemical interactions, which is the rate-limiting step that controls the adsorption process.

Using magnetic graphene in the removal of malachite green [13], polyethyleneimine magnetic nanoadsorbents in the elimination of methyl orange and Pb(II) [33], magnetic cyclodextrins in the removal of Eu(III) [34], and magnetic β-cyclodextrin porous polymer nanospheres in the uptake or organic pollutants [35], similar kinetics were observed.

To understand the adsorption of Direct Red 83:1 onto magnetic polymers, the kinetics of the adsorption process were analyzed by means of the intraparticle diffusion model to determine whether intraparticle diffusion occurs by plating and to determine whether it plays an essential role in the adsorption process. This effect was studied by plotting the amount of Direct Red 83:1 dye absorbed versus the square root of time [36].

Figure 11C shows two steps in the adsorption of Direct Red 83:1: the first straight region is associated with chemisorption, and the second region is controlled by the intraparticle diffusion model (effect of the boundary layer).

### 2.6. Isotherm Analysis

The interaction between dyes and adsorbents is explained by means of adsorption isotherms and gives an idea of the adsorption capabilities of the adsorbent [37]; the results are presented in Table 2. The Freundlich model describes adsorption processes onto heterogeneous surfaces, with energetically different adsorption sites, and essentially yields the constants K_F_ and n_F_ [36]. When adsorption is favorable, n_F_ ranges from 1 to 10. A higher result of nF determines that the interaction between the dye and polymer is adequate.

The K_F_ value for the β-CDs-EPI-Fe polymer was 0.19 L/g. According to the n_F_ results, it was higher than 1; therefore, the adsorption trend was approximately linear and basically favored (1.04). The adjustment of the experimental data to the Freundlich model suggests that the CDs-EPI-Fe polymer followed this isotherm, taking into consideration the value of R^2^ (0.911). According to these results, it is possible to conclude that adsorption occurs on heterogeneous surfaces in this polymer (Figure 12A).

On the other hand, the Langmuir model was adequate in the case of materials presenting uniformly energetic adsorption sites and monolayer adsorbate coverage, and the isotherm assumed that all of the sites were equivalent to uniform surface coverage [38].

The analysis of q_max_ is necessary because it indicates the maximum adsorption under the experimental conditions assayed. In the case of the β-CD-EPI-Fe polymer, the qmax value was 32 mg/g. The analysis of the separation factor (R_L_) showed that the results for this parameter ranged from 0 to 1, indicating that this process is favorable and confirm the results observed in the analysis of the n_F_ parameter from the Freundlich model (Figure 12B).

Finally, the experimental data were adjusted to the Tempkin isotherm. Furthermore, this isotherm is valid when the adsorption heat linearly decreases as a function of the coverage degree because of the interactions between the substrate and the adsorbent. In this case, the process is characterized by a uniform distribution of binding energies [39]. The binding energies ranged from 8–16 kJ/mol in the case of ionic exchange and from −40 kJ/mol in physical adsorption [40]. The b_T_ value obtained for the CD-EPI-Fe polymer was 0.37 kJ/mol.

According to this result, the physical and chemical forces were both involved in the adsorption of Direct Red onto the magnetic adsorbent (Figure 12C).

Our research group has extensively analyzed the elimination of Direct Red 83:1 by means of different adsorbents, and the efficiency in the removal of this dye can be observed in Table 3.

The adsorption capacity of the β-CDs-EPI-Fe polymer was similar or higher to the rest of the polymers except for that of β-CDs-EPI. Both polymers had a common structure but the β-CDs-EPI-Fe polymer had iron added to it and this feature affected its efficiency. The decrease of adsorption capacity caused by the incorporation of iron is a comparative disadvantage, but it is the toll to be paid to have a polymer magnetically separable. The polymer exhibited excellent reusability properties; after six cycles of loading and desorption, its absorption capacity was still 90% (Appendix A).

### 2.7. Thermodynamic Parameters

The Gibbs free energy value (Δ*G*°) is related to the spontaneity of chemical reactions. To determine this value, Equation (1) was used:(1)$$K^{{}^{\circ}}=K_{p}\times M_{adsorbate}\times 55.5$$
where *K_p_* is the equilibrium constant (L/g); *M_adsorbate_* is the molecular weight of Direct Red; and 55.5 is the constant related to the mole concentration of water (mol/L) [43,44]. The results from Equation (1) were used in Equation (2).
(2)$$\Delta G^{{}^{\circ}}=-RTln\;lnK^{{}^{\circ}}\;$$

The standard free energy (Δ*G*°) was −27,556.9 J/mol for β-CD-EPI at room temperature, confirming the spontaneity of the adsorption process. Enthalpy (Δ*H*°) and entropy (Δ*S*°) were calculated by using the Van’t Hoff plot (data not shown). At room temperature, Δ*H*° was 8757.1 J/mol (endothermic reaction) and Δ*S*° was 122.4 J/mol.

As pointed out by Saha and Chowdhury (2011) [45], a value of Δ*S*° > 0 indicates increased randomness at the solid/solution interface; the adsorbed solvent molecules gain more translational entropy than is lost by the adsorbate ions/molecules and the degree of freedom of the adsorbed species increase.

### 2.8. Advanced Oxidation Process

The degradation of Direct Red 83:1 by the pulsed light/H_2_O_2_ process is shown in Figure 13.

It can be observed that decoloration of the assay mixture becomes slower with the progress of the treatment, which is typical of pseudo-first-order kinetics, exhibiting a degradation rate of 0.0196 cm^2^/J (R^2^ = 0.9947). More than 90% of the dye is degraded within the first 55 pulses (117.7 J/cm^2^). Prolonging the treatment is not efficient, and data extrapolation predicts requiring doubling the fluence to reach 99% degradation.

The novel β-CD-EPI magnetic polymer is small, with a high surface area to volume ratio and low diffusion resistance, which favors the adsorption kinetic. Its magnetic properties make it easily separable from water. In addition, the entrapped dye can be desorbed and reused by at least six cycles without significantly losing capability (>90%), which allows for the reuse of both the dye and water, in a new dying cycle. Furthermore, the polymer can also be reused in a new dye removal round, which is in harmony with the circular economy concept. Coupling this novel polymer with a novel AOP offers a fast and efficient way of minimizing water pollution in the dyeing industry.

## 3. Materials and Methods

### 3.1. Chemicals and Reagents

Commercially available β-CD was supplied by Arachem (Tilburgo, The Netherlands). Sodium borohydride (98%), sodium hydroxide (98%), epichlorohydrin (99%), iron(III) chloride hexahydrate (FeCl_3_.6H_2_O), ethanol and acetone were purchased from Sigma-Aldrich (Madrid, Spain). Iron(II) chloride tetrahydrate (FeCl_2_.4H_2_O) and ammonium hydroxide solution were supplied by Fluka (Madrid, Spain). Direct Red 83:1 (CAS number: 90880-77-6) was supplied by AITEX (Asociación de Investigación de la Industria Textil, Alcoy, Spain).

### 3.2. Iron Nanoparticles Preparation

First, 3.49 g of FeCl_2_·4H_2_O and 9.5 g of FeCl_3_·6H_2_O were mixed in 100 mL water at 25 °C, and then 30 mL of ammonium hydroxide was added dropwise. After magnetic stirring for 30 min at 80 °C, the mixture was centrifuged at 4000 rpm for 10 min. The precipitate was washed with ethanol:water (1:1), the washing step was repeated twice, and then the precipitate was dried overnight to obtain magnetic nanoparticles.

### 3.3. Epichlorohydrin-Iron-β-Cyclodextrin Polymer Preparation

The β-CD-EPI-magnetic (β-Cyclodextrin-Epichlorohydrin) polymer was produced using the protocol described by Pellicer et al. (2018) [41], with slight modifications to obtain a magnetic polymer. First, 60 mg of sodium borohydride, 24 g of β-CDs, 24 g of Fe nanoparticles and 24 mL of water were stirred for 10 min at 50 °C. After this, 26 mL of sodium hydroxide (40%) was poured and stirred for 5 min. Once that, 264 g of EPI was added dropwise. The mixture was stirred for 6 h at 50 °C to obtain the adsorbent. The polymer was washed with water:acetone (2:1) and dried overnight at 60 °C.

### 3.4. Characterization of the Polymer Material

The characterization of the β-CD-EPI polymer-modified Fe_3_O_4_ beads was accomplished by various methods. Carbon, hydrogen, nitrogen and sulfur elemental compositions were determined using a LECO CHNS-932 analyzer (LECO Instruments, St. Joseph, MI, USA) with a threshold value of 0.2% for each element. The particle size study was performed by water dispersion in the Hydro 2000G unit of the Mastersizer 2000LF laser analyzer (Malvern Instruments Ltd., Worcestershire, UK), covering a measurement range of 0.02–2000 µm, whereas the specific surface area was determined in accordance with Mie theory by the light scattered from the particles taken as equivalent spheres. The morphology was resolved by the analysis of FE-SEM images provided by an Apreo S field emission scanning electron microanalyzer equipped with an EDX detector (Thermo Scientific Brno, Brno, Czech Republic). The energy dispersive X-ray spectroscopy mapping was performed at an accelerating voltage of 20 kV.

IR spectra measurements of the samples were carried out on a Nicolet 5700 spectrometer (Nicolet, Madison, WI, USA) equipped with a Ge/KBr beam splitter, a DTGS-KBr detector and a ceramic infrared source over the range 4000 to 400 cm^−1^, 4 cm^−1^ of resolution. XRD measurements were performed on a Bruker D8 Advance diffractometer (Bruker Corporation, Billerica, MA, USA) by Cu-Kα radiation and scanned at 40 kV and 30 mA from 10° to 70° in the 2θ range, a step size of 0.05° steps, 1 s/step and an angular velocity of 30 rpm. The powder diffraction file was evaluated with linked software (DIFFRAC. EVA 5.2, Bruker AXS, 2020) and a crystalline powder database (PDF-4+ 2021, ICDD). TGA and DSC were performed on Mettler-Toledo Instruments DSC 822e and TGA/DSC-1 HT (Schwerzenbach, Switzerland), respectively, under a dynamic atmosphere of nitrogen. Thermal scans were performed from 30 to 800 °C (up to 450 °C for DSC) at a heating rate of 10 °C/min. For differential thermal analysis (DTA), thermobalance was coupled to a Balzers Thermostar mass spectrometer (Pfeiffer Vacuum, Asslar, Germany) for gas analysis. Scan bar graph cycles were performed in the range 15–94 *m/z* in the quadrupole mass spectrometer (QMS 200 M3), with a dwell time of 2 s for every ion and cathode voltage in the ion source of 65 V.

### 3.5. Dye Solution Preparation

To achieve adsorption experiments, Direct Red 83:1 (CAS 90880-77-6, molecular weight of 992.77 g/mol) was initially prepared at several concentrations (50, 100, 150, 200 and 300 mg/L). The remaining dye concentration was measured in the supernatant using a colorimeter (Shimadzu UV-1603). Absorbance signatures were monitored upon treatment at the maximum absorbance of this dye (λmax = 526 nm; ε_526_ = 1065 M^−1^ cm^−1^).

### 3.6. Adsorption Experiments

Adsorption tests were performed at 25 °C using solutions containing different concentrations of dye (50 to 300 mg/L). In each experiment, 0.5 g of polymer and 50 mL of dye solution (pH = 5) were mixed. The mixture was stirred at 500 rpm. The amount of dye not retained in the polymer was measured every 10 min; to obtain this remaining dye, external magnets were used for 5 min. Then, the concentration of dye was determined by colorimetric experiments. All experiments were performed in triplicate.

The quantity of dye entrapped on the polymer (*q_e_*) was determined following Equation (3) [46]:(3)$$q_{e}=\frac{V\left(C_{0}-C_{e}\right)}{m}$$
where V is the volume of dye (L), m is the mass of polymer employed (g), *C*_0_ stands for the concentration in the liquid phase at the earliest stage (mg/L), and *C_e_* indicates the liquid phase dye concentration at equilibrium (mg/L). All experiments were conducted in triplicate.

### 3.7. Kinetics and Isotherm Analysis

Aiming to assess the mechanism that governs the adsorption of the dye, the pseudo-first-order [47], pseudo-second-order [48,49], and intraparticle diffusion models [50] were evaluated using Equations (4)–(6), respectively.
(4)$$\log(q_{e}-q_{t})=logq_{e}-\frac{k_{1}}{2.303}t$$
(5)$$\frac{t}{q_{t}}=\frac{1}{k2qe^{2}}+\frac{1}{q_{e}}\;t$$
(6)$$q_{t}=ki\sqrt{t}+C$$
where *q_e_* and *q_t_* are the amounts of dye adsorbed (mg/g) at equilibrium and at time t (min), respectively; *k*_1_ indicates the pseudo-first-order rate constant (min^−1^); *k*_2_ represents the equilibrium rate constant of the pseudo-second-order (g/mg min); *k_i_* is the intraparticle diffusion rate constant (mg/g min^1/2^); *t* is the time; and *C* is the intercept (mg/g).

Adsorption isotherms provide useful information to elucidate the way in which molecules are distributed between the solid and liquid phases if the equilibrium time is reached. In this study, three conventional equations, namely, the Freundlich [51], Langmuir [52], and Tempkin [53], were investigated to find the best-fitted model to understand the adsorption of Direct Red onto the β-CD-EPI magnetic polymers. The goodness of fit model is expressed by the determination coefficients (R^2^). The Freundlich, Langmuir, and Tempkin equations are given in Equations (7)–(9), respectively:(7)$$ln\;q_{e}=ln\;K_{F}+\frac{1}{n_{F}}\;ln\;C_{e}$$
(8)$$\frac{Ce}{q_{e}}=\frac{1}{K_{L}}+\frac{a_{L}}{K_{L}}C_{e}$$
(9)$$RL=\frac{1}{1+a_{L}C_{0}}$$
(10)$$q_{e}=\frac{RT}{b_{T}}\ln a_{T}+\frac{RT}{b_{T}}\ln C_{e}$$
where *q_e_* is the equilibrium dye concentration on the adsorbent (mg/g); *C_e_* is the equilibrium dye concentration in solution (mg/L); *K_F_* is the Freundlich constant (L/g); 1/*n_F_* is the heterogeneity factor; *K_L_* (L/g) and *a_L_* (L/mg) are the Langmuir isotherm constants; *R_L_* is the separation factor; *b_T_* is the Tempkin constant (kJ/mol); *a_T_* is the constant of the Tempkin isotherm (L/g); *R* is the universal gas constant (8.314 J/mol K); and *T* is the absolute temperature in Kelvin. The value of *R_L_* indicates the isotherm type as irreversible (*R_L_
*= 0), favorable (0 < *R_L_* < 1), linear (*R_L_* = 1), or unfavorable (*R_L_
*> 1).

### 3.8. Polymer Reusability

The reusability of the EPI-β-CDs-Fe polymer was evaluated using the same dye at 50 mg/L. A total of 50 mL of dye solution was mixed with 1 g of polymer and stirred during 1 h at 500 rpm. Then, the polymer was magnetically separated for 10 min and the remaining concentration of dye was determined spectrophotometrically, as above-mentioned. The dye solution was decanted and the separated polymer was then regenerated using 50 mL acetate buffer, pH 4, 220 mM, for 30 min. Thereafter, the polymer was once more magnetically separated and loaded with dye for a new use cycle up to six.

### 3.9. Dye Degradation by Pulsed Light/H_2_O_2_ Process

The elimination of Direct Red 83:1 by the pulsed light/H_2_O_2_ process was assayed to further decrease the contamination level that could eventually reach the environment. The approach simulates a two-step sequential process where the water contaminated by the dye is first treated with the polymer and then this water, containing the amount of dye that is not adsorbed by the polymer, is treated by the pulsed light/H_2_O_2_ process. To this end, a 20 mL mixture of dye and H_2_O_2_ was prepared at final concentrations of 45 mg/L dye and 343 mg/L H_2_O_2_. The dye was prepared at the concentration that left the polymer, and the concentration of H_2_O_2_ was 200 times greater than that of the dye on a molar basis to have an excess of H_2_O_2_, which avoids making it the limiting reagent. This mixture was treated with 60 light pulses, each amounting to 2.14 J/cm^2^ measured at the mixture surface level for a final fluence of 128.4 J/cm^2^. Pulsed light treatment was performed with a XeMaticA-Basic-1 L (Steribeam, Kehl, Germany) operated at 2.5 kV, and the emission spectrum was reported [54]. Samples were withdrawn every five pulses to measure absorbance. Experiments were performed in duplicate. Data were normalized and fitted using pseudo-first-order kinetics.

### 3.10. Molecular Models

The structure of Direct Red 83:1 was initially retrieved from PubChem (database available at https://pubchem.ncbi.nlm.nih.gov/compound/101609554, accessed on 22 June 2022). The deposited Cartesian coordinates were next implemented in the Schrödinger suite of programs (Schrödinger Release 2021-1, Schrödinger, LCC, New York, NY, USA, 2021), which was used to generate stable 3D structures at all selected pH values. All protonation states were determined with the Epik code [55,56]. This approach is known to correctly reproduce the pKa values in azo dyes [57]. All resulting molecules were subsequently docked into the central cavity of a β-CD [58] by using the extra precision version of Glide as a docking engine [59]. A final refinement was conducted with Prime [60]. The latter level of theory accounts for β-CD relaxation upon encapsulation and consequently yields more accurate energy values.

## 4. Conclusions

The removal of Direct Red 83:1 from wastewater was achieved by using β-CD-EPI magnetic adsorbents. An acidic environment (0.5 g) was the optimum condition to perform the whole set of experiments. The experimental data followed the pseudo-second and intraparticle diffusion models. Adsorption occurred onto heterogeneous surfaces according to the three isotherms analyzed. The magnetic adsorbent was able to remove 32 mg/g of Direct Red 83:1 (q_max_). The adsorption was exergonic according to the Gibbs free energy results, which indicates the spontaneity of this adsorption process. The novel β-CD-EPI magnetic polymer is small, with a high surface area to volume ratio and low diffusion resistance, which favors the adsorption kinetic. Its magnetic properties make it easily separable from water. The polymer exhibited a higher reusability, keeping 90% of its capacity after six cycles of loading–desorption, which is in harmony with the circular economy concept. The remaining dye in solution after polymer treatment was successfully eliminated using an advanced oxidation process. Coupling this novel polymer with a novel AOP offers a fast and efficient way of minimizing water pollution in the dyeing industry.

## Figures and Tables

**Figure 1 ijms-23-08406-f001:**
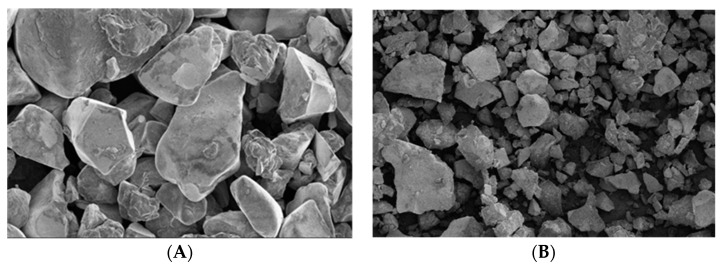
(**A**) Surface morphology structure of the β-CD-EPI polymer-modified Fe_3_O_4_ nanoparticles (2.0 kV; magnification 100×). (**B**) Surface morphology and structure of the powder of the β-CD-EPI polymer-modified Fe_3_O_4_ nanoparticles (2.0 kV; magnification 150×).

**Figure 2 ijms-23-08406-f002:**
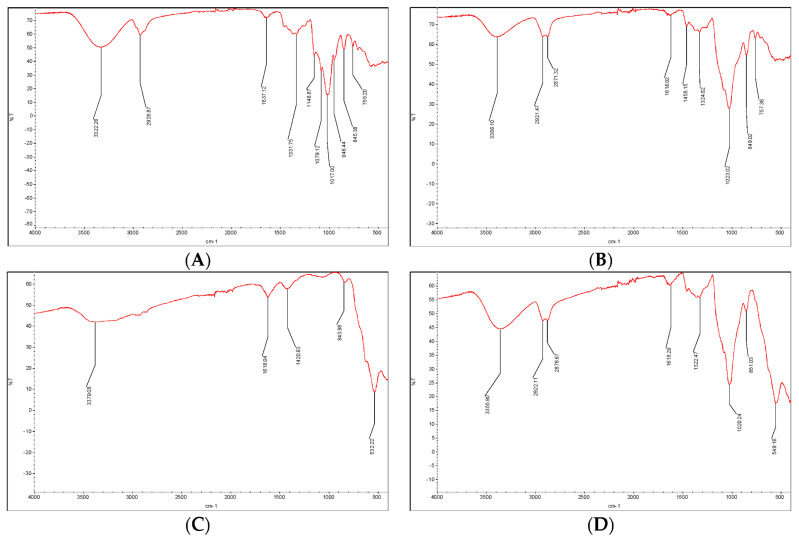
The IR spectra of the (**A**) β-cyclodextrin, (**B**) β-cyclodextrin-epichlorohydrin, (**C**) Fe_3_O_4_ nanoparticles, and (**D**) β-CD-EPI polymer-modified Fe_3_O_4_ nanoparticles.

**Figure 3 ijms-23-08406-f003:**
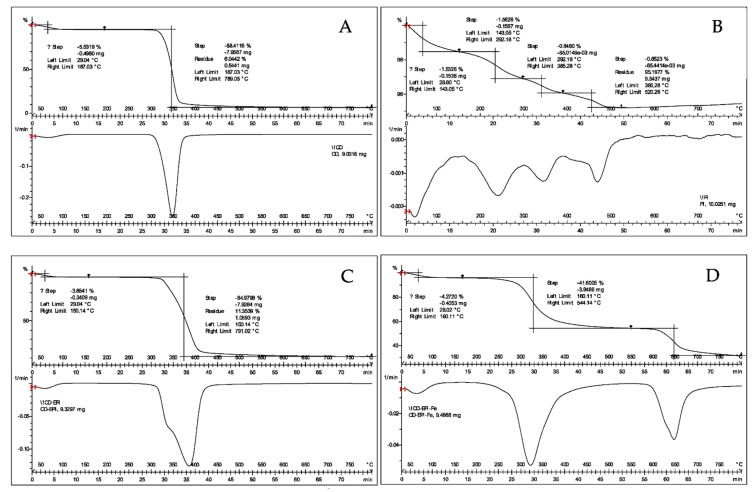
The differential thermal analysis (DTA) curves for the (**A**) β-cyclodextrin, (**B**) Fe_3_O_4_ nanoparticles, (**C**) β-cyclodextrin-epichlorohydrin polymer, and (**D**) β-CD-EPI polymer-modified Fe_3_O_4_ nanoparticles.

**Figure 4 ijms-23-08406-f004:**
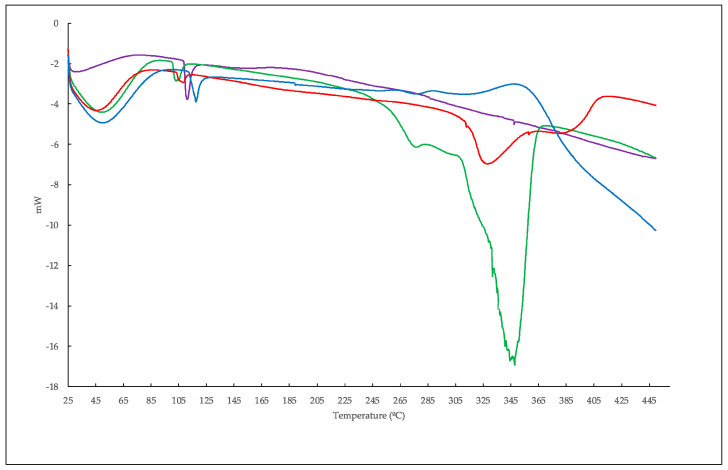
The differential scanning calorimetry (DSC) curves for the (green) β-cyclodextrin, (lilac) Fe_3_O_4_ nanoparticles, (red) β-cyclodextrin-epichlorohydrin polymer, and (blue) β-CD-EPI polymer-modified Fe_3_O_4_ nanoparticles.

**Figure 5 ijms-23-08406-f005:**
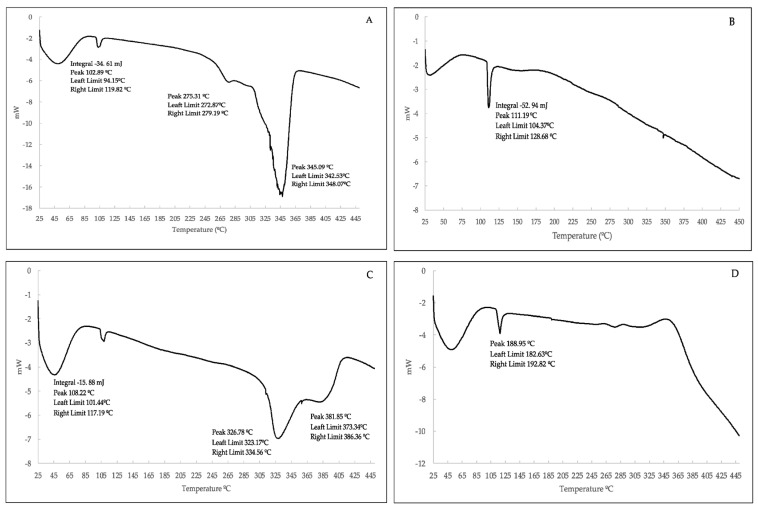
The differential scanning calorimetry (DSC) curves for the (**A**) β-cyclodextrin, (**B**) Fe_3_O_4_ nanoparticles, (**C**) β-cyclodextrin-epichlorohydrin polymer, and (**D**) β-CD-EPI polymer-modified Fe_3_O_4_ nanoparticles.

**Figure 6 ijms-23-08406-f006:**
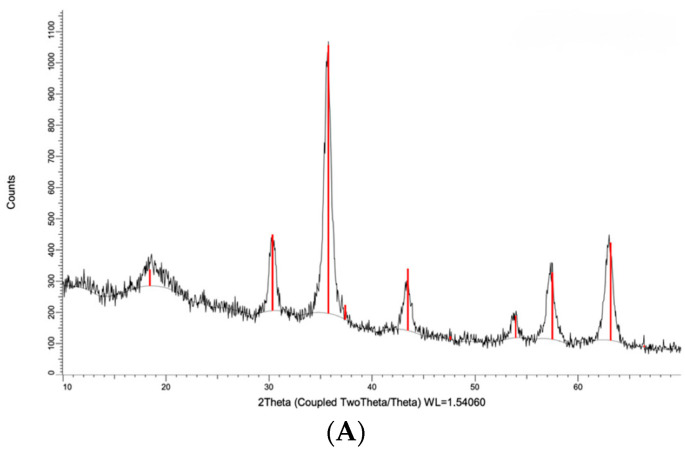
(**A**) The XRD patterns of the β-CD-EPI polymer-modified Fe_3_O_4_ nanoparticles, (**B**) Fe_3_O_4_ nanoparticles.

**Figure 7 ijms-23-08406-f007:**
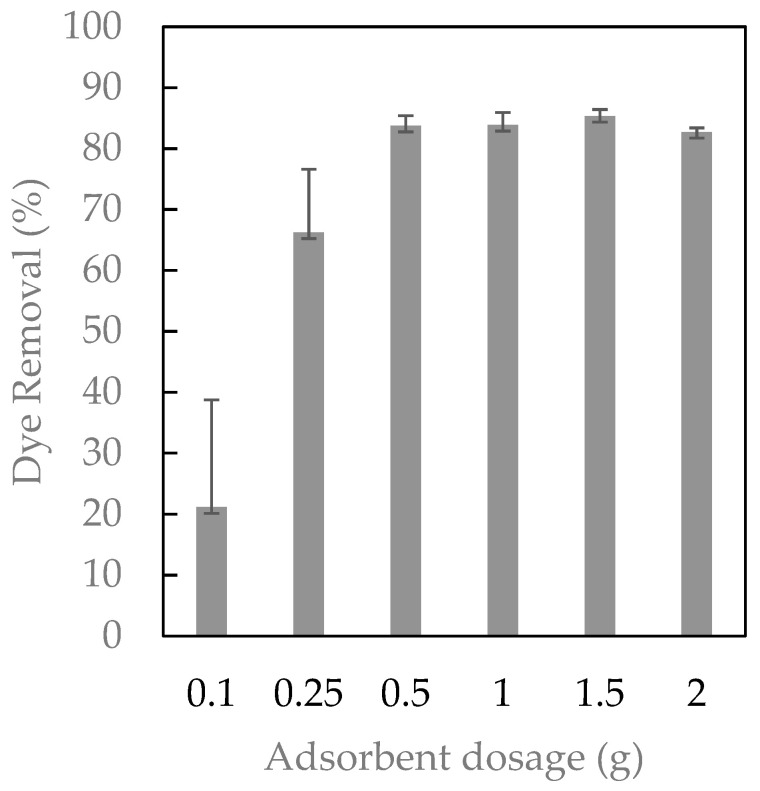
The effect of adsorbent dosage on the removal of Direct Red 83:1.

**Figure 8 ijms-23-08406-f008:**
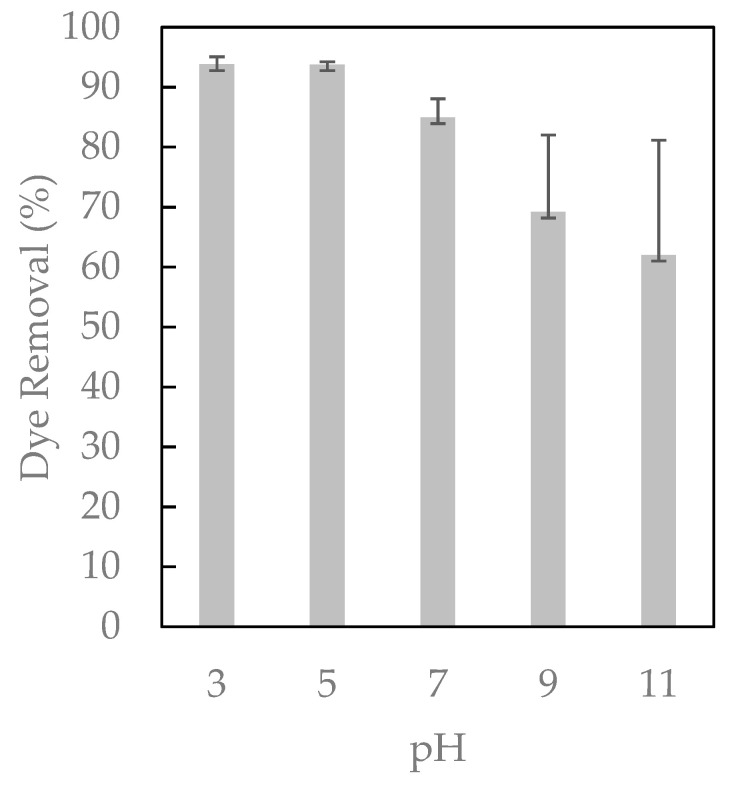
The effect of pH in the removal of Direct Red 83:1.

**Figure 9 ijms-23-08406-f009:**
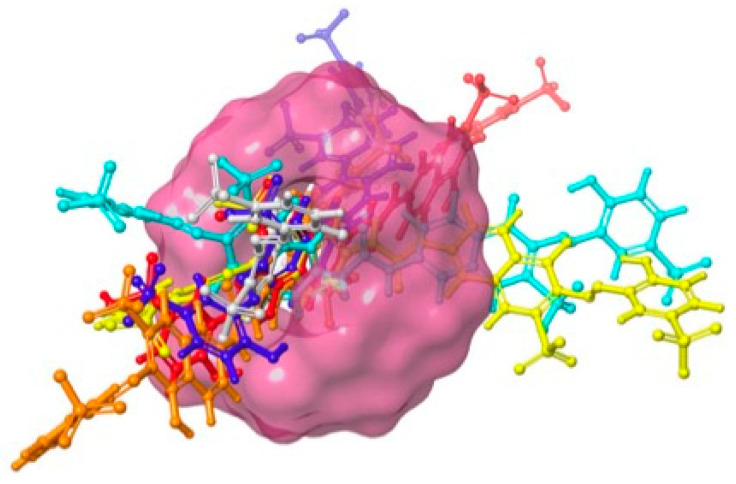
A summary of the results by using computational methods. The β-CD is displayed as a pink surface while Direct Red 83:1 is represented with ball and stick models. Color scheme: red, charge = −3; orange, charge = −4; yellow, charge = −5; cyan, charge = −6; blue, charge = −7; grey, charge = −8.

**Figure 10 ijms-23-08406-f010:**
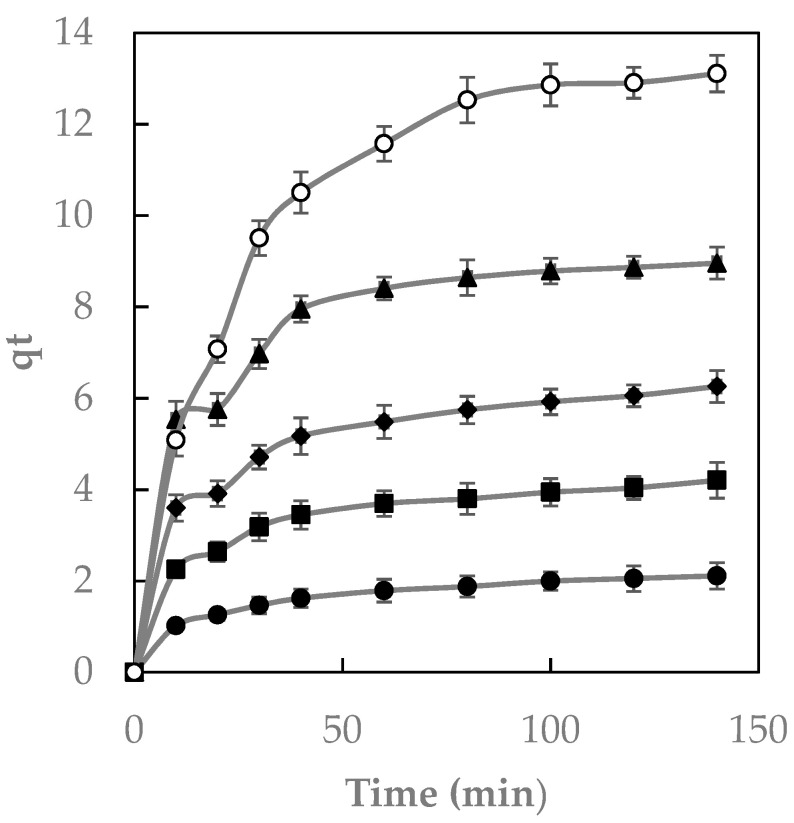
The effect of contact time at different dye concentrations: 50 (●), 100 (■), 150 (◆), 200 (▲), and 300 (○) mg/L. N = 3.

**Figure 11 ijms-23-08406-f011:**
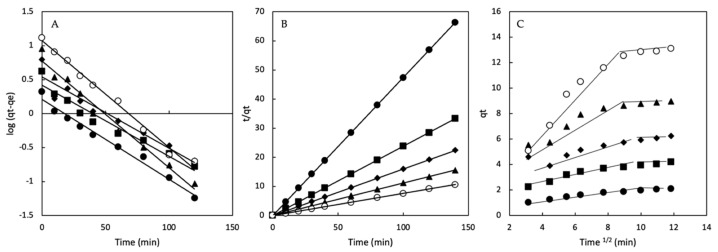
The kinetics analysis (**A**) pseudo-first model; (**B**) pseudo-second model; (**C**) intraparticle diffusion model. Co: 50 (●), 100 (■), 150 (◆), 200 (▲), and 300 (○) mg/L. N = 3.

**Figure 12 ijms-23-08406-f012:**
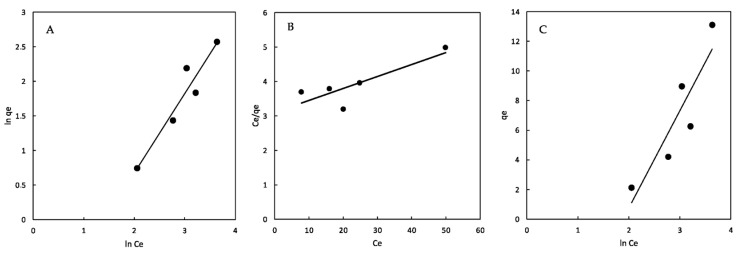
The isotherm analysis. (**A**) Freundlich model; (**B**) Langmuir model; (**C**) Tempkin model.

**Figure 13 ijms-23-08406-f013:**
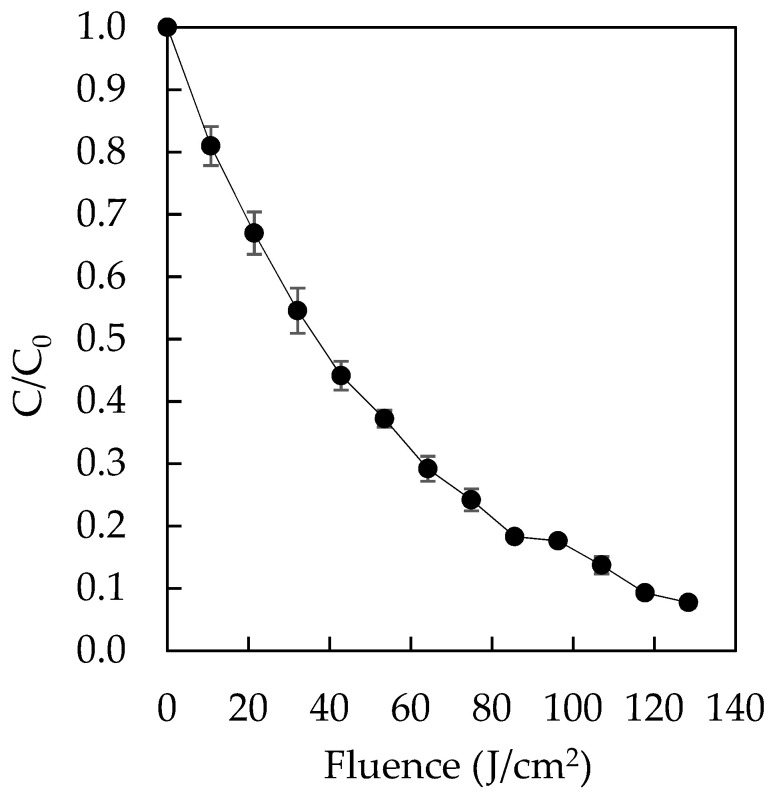
The degradation of residual Direct Red 83:1 in solution by means of pulsed light/H_2_O_2_.

**Table 1 ijms-23-08406-t001:** The kinetic results (pseudo-first, pseudo-second, and intraparticle diffusion models) for the β-CD-EPI-Fe polymer.

**PFOM**	**β-CDs-EPI Magnetic Polymer**			
*Co* (mg/L)	*qe_exp_*	*qe_cal_*	*K* _1_ ^(min−1)^	R^2^
50	2.10	1.61	0.027	0.982
100	4.20	2.60	0.024	0.950
150	6.26	3.48	0.024	0.924
200	8.95	5.93	0.036	0.978
300	13.10	11.78	0.036	0.989
**PSOM**	**β-CDs-EPI Magnetic Polymer**			
*Co* (mg/L)	*qe_exp_*	*qe_cal_*	*K* _2_ ^(g/mg min)^	R^2^
50	2.10	2.10	0.225	0.999
100	4.20	4.20	0.168	0.999
150	6.26	6.26	0.105	0.999
200	8.95	8.62	0.035	0.999
300	13.10	13.10	0.013	0.999
**IDM**	**β-CDs-EPI Magnetic Polymer**			
*Co* (mg/L)	*qe_exp_*	*qe_cal_*	*K_i_* ^(mg/g min 1/2)^	R^2^
50	2.10	0.74	0.120	0.959
100	4.20	1.85	0.21	0.920
150	6.26	3.46	0.24	0.872
200	8.95	4.58	0.41	0.861
300	13.10	3.75	0.89	0.878

**Table 2 ijms-23-08406-t002:** The isotherm results for the Freundlich, Langmuir, and Tempkin models.

Isotherm	Parameter	CDs
Freundlich	K_F_	0.190
	n_F_	1.040
	R^2^	0.911
Langmuir	q_max_	32
	K_L_	0.320
	a_L_	0.010
	R^2^	0.747
	R_L_	0.667–0.250
Tempkin	a_T_	0.150
	b_T_	0.370
	R^2^	0.807

**Table 3 ijms-23-08406-t003:** The adsorption efficiency of different polymers to remove Direct Red 83:1.

Adsorbent	q_max_ (mg/g)	Reference
α-CDs-EPI	31.5	[41]
HP-α-CDs-EPI	23.4	
β-CDs-EPI	107.5	[42]
HP-β-CDs-EPI	18.2	
γ-CDs-EPI	11.9	[39]
HP-γ-CDs-EPI	14.6	
β-CDs-EPI-Fe	32.0	This work

## Data Availability

Not applicable.

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
