# Peer review of "Removal of an Azo Dye from Wastewater through the Use of Two Technologies: Magnetic Cyclodextrin Polymers and Pulsed Light"

_ijms, 2022, doi:10.3390/ijms23158406_

Round 1
Reviewer 1 Report
Comments and Suggestions for Authors
The paper deals with the synthesis and characterisation of cyclodextrin polymers decorated with magnetic nanoparticles that can act as means to remove and separate dyes from wastewater. I consider it appropriate to be published in Int. J. Mol. Sci. after the following major revisions:
- More references referring to magnetic cyclodextrin polymers are needed. Any updated bibliographic citations that refer the potential applications of these systems would be advisable.
- Figure 1 (FT-IR) needs some resolution adjustments, as it is difficult to analyse the spectra as shown. The same goes for Figure 2 (TGA).
- FE-SEM analyses should be included in the main manuscript, as pore size is a crucial parameter when it comes to pollutants removal from wastewater.
- XRPD of the polymer and the Magnetite-Polymer system should be compared in the same diffractogram rather than the former being in the Supplementary Materials.
- There is no information about the magnetic response or magnetization saturation (a vibrating sample magnetometer will do) for the magnetic nanoparticles nor the magnetite-polymer systems.
- The removal of the dye with the polymers and magnetic adsorbents (Table 3) should be compared with native cyclodextrins as a blank.
- DSC data should be included in the main manuscript to support the discussion in the TGA results.
Author Response
The paper deals with the synthesis and characterisation of cyclodextrin polymers decorated with magnetic nanoparticles that can act as means to remove and separate dyes from wastewater. I consider it appropriate to be published in Int. J. Mol. Sci. after the following major revisions:
Dear reviewer, thank you very much for your valuable comments with the aim to improve the present manuscript.
- More references referring to magnetic cyclodextrin polymers are needed. Any updated bibliographic citations that refer the potential applications of these systems would be advisable.
Thanks for your recommendation. Various references have been included in the introduction and they were highlighted in yellow.
2. Figure 1 (FT-IR) needs some resolution adjustments, as it is difficult to analyse the spectra as shown. The same goes for Figure 2 (TGA).
Thanks for your recommendation. We have analyzed the overall quality of our figures and we have improved them accordingly.
3. FE-SEM analyses should be included in the main manuscript, as pore size is a crucial parameter when it comes to pollutants removal from wastewater.
Thanks for your recommendation. FE-SEM analysis has been incorporated into the manuscript. During SEM analysis we tried to increase the resolution to be able to appreciate the pore size in the particles through image analysis. However, being an iron oxide crystal of about 82.7 A (XRD, reflection 311, data not shown) coated by amorphous organic matter, when we tried to increase the resolution, the electron beam energy heated up significantly the sample forcing us to stop the diffraction before obtaining accurate data.
In any case, although we recognize that it is an important parameter, the ability of removing contaminants, in particular the one we have tested in this article, has been demonstrated regardless of the pore size of the particles of the polymer.
4. The same diffractogram rather than the former being in the Supplementary Materials.
Thanks for your recommendation. The images of the supplementary material have been incorporated into the discussion of the manuscript.
5. There is no information about the magnetic response or magnetization saturation (a vibrating sample magnetometer will do) for the magnetic nanoparticles nor the magnetite-polymer systems.
Thanks for your recommendation. We have to apologize for the lack of a magnetic response for our polymer. When we produced the polymer we did not have access to a magnometer. We were looking for external analysis service but we were not able to locate anyone to perform the analysis for us. However, we are considering some improvements in the polymer for the following articles and then we will try again to fully characterize it, including the magnetic response.
6. The removal of the dye with the polymers and magnetic adsorbents (Table 3) should be compared with native cyclodextrins as a blank.
Thanks for your comment. Cyclodextrins are water-soluble molecules like the dye, direct red 83:1. It is necessary to develop insoluble cyclodextrin polymers in order to remove the dye from solution. After performing this reaction we are able to measure the remaining dye in solution. For this reason, the comparison with the native cyclodextrin has not been made, since we need the use of polymers to be able to eliminate the dye from the residual water. However, in our previous articles the synthesis of insoluble CD polymers without magnetic properties were developed and the comparison is observed in Table 3.
7. DSC data should be included in the main manuscript to support the discussion in the TGA results.
Thanks for your advice. DSC analysis has been incorporated into the manuscript.

Reviewer 2 Report
Good job; the only issue I found was the figures; all the figures were unclear and hard to read the subtle notes. Increasing the size of the figures or making them more clear would be suitable.
Author Response
Good job; the only issue I found was the figures; all the figures were unclear and hard to read the subtle notes. Increasing the size of the figures or making them more clear would be suitable.
Thanks for your recommendation and for your nice feedback of our article. We have analyzed the overall quality of our figures and we have improved them accordingly.

Reviewer 3 Report
The authors have submitted an interesting article with the title “Effective removal of an azo dye from wastewater using a sequential process adsorption in a cyclodextrin magnetic polymer and trace degradation by pulsed light”. I recommend the publication of this work after a major revision.
1) Author should revise the title since it is too long (24 words). It is recommended that the title should not exceed 17 words.
2) In lines 17-18, the authors mentioned “there is a necessity to produce new decolorization methods …”. However this work does not present a new technique for wastewater treatment rather than a new composite.
3) Some sentences are not clear and need to be revised. For example:
a) “The equilibrium contact time was 30 min with a 26 qmax of 32.0 mg/g” in line 26.
b) “The newly synthetized β-CD-EPI-magnetic polymer exhibited good adsorption properties and separability from water which, when complemented with a pulsed light-AOP, may offer a good alternative to remove dyes such as Direct Red 83:1 from water and reuse both, dye 40 and water, in the dying process” in lines 38-41.
4) In the introduction part, there are several sentences without references. Avoid adding all the references at the end of the paragraph. It is recommended to cite every sentence individually to be more accessible to readers. For example lines 60-70.
5) Be consistent in using abbreviations. For example, FE-SEM was used in line 23 and again repeated “field emission scanning electron microscope” in lines 140 and 434. A similar situation happens for XRD and others. Revise them in the manuscript.
6) Quality of the presented graphs is low. Try using GraphPad or OriginPro to provide high-quality images.
7) In the kinetic experiment, the authors mentioned the number of repetitions was 3 (N=3). However, the graph misses the error bar.
8) In some sentences authors used “absorbent’ and “absorption” which need to be revised. For example: Lines 174, 247, 251, etc.
9) Some graphs such as Figures 4 and 5, show error bars that need to mention the reputation number or change the graph to the “scatter plot with error bar”.
10) It is recommended to do a comparison study or provide a table and compare the recent azo dye removal capacity of adsorbent with your work to emphasize the advantages of your work. You can use similar studies like the following refs. 10.1016/j.jhazmat.2019.01.107 and 10.1016/j.molliq.2018.12.050
11) Synthesis process, adsorption, and further photodegradation can be illustrated as a graphical abstract to increase the impact of the work. Same references as comment number 10 can help you to come up with a good idea.
Pay attention to connectors and revised the whole manuscript to improve its English of it. And correct the typos. For example “synthetized” in line 38 should change to “synthesized”
Author Response
The authors have submitted an interesting article with the title “Effective removal of an azo dye from wastewater using a sequential process adsorption in a cyclodextrin magnetic polymer and trace degradation by pulsed light”. I recommend the publication of this work after a major revision.
Thank you very much for your good feedback.
- Author should revise the title since it is too long (24 words). It is recommended that the title should not exceed 17 words.
Thanks for your suggestion. The title has been updated.
- In lines 17-18, the authors mentioned “there is a necessity to produce new decolorization methods …”. However this work does not present a new technique for wastewater treatment rather than a new composite.
Thanks for your recommendation. The sentence has been reformulated and highlighted in the new version of the manuscript.
- Some sentences are not clear and need to be revised. For example:
a) “The equilibrium contact time was 30 min with a qmax of 32.0 mg/g” in line 26.
b) “The newly synthetized β-CD-EPI-magnetic polymer exhibited good adsorption properties and separability from water which, when complemented with a pulsed light-AOP, may offer a good alternative to remove dyes such as Direct Red 83:1 from water and reuse both, dye 40 and water, in the dying process” in lines 38-41.
Thanks for your recommendation. Sentences have been rephrased to increase the clarity of them and highlighted in yellow in the new version of the manuscript.
- In the introduction part, there are several sentences without references. Avoid adding all the references at the end of the paragraph. It is recommended to cite every sentence individually to be more accessible to readers. For example lines 60-70.
Thanks for your valuable advice. Various references have been included in the introduction section and again highlighted in yellow.
- Be consistent in using abbreviations. For example, FE-SEM was used in line 23 and again repeated “field emission scanning electron microscope” in lines 140 and 434. A similar situation happens for XRD and others. Revise them in the manuscript.
Thanks for your suggestion. We have checked the use of abbreviations throughout the manuscript.
- Quality of the presented graphs is low. Try using GraphPad or OriginPro to provide high-quality images.
Thanks for your recommendation. We have analyzed the overall quality of our figures and we have improved them accordingly.
- In the kinetic experiment, the authors mentioned the number of repetitions was 3 (N=3). However, the graph misses the error bar.
In the case of kinetic analysis, the goodness of PFOM, PSOM and IDM is evaluated according to the determination coefficient (R2) and the similarities between calculated and experimental values. In addition, the comparison between adsorbents is purely based on some experimental results, it is especially useful the result of qmax. Again here, the result of R2 allowed us to establish some conclusions regarding the type of interaction between dye and polymer. According to our humble point of view, the addition of error bars to Figure 11 increases the difficulty to comprehend the results. For these mentioned reasons, error bars were not included in Figure 11.
- In some sentences authors used “absorbent’ and “absorption” which need to be revised. For example: Lines 174, 247, 251, etc.
Thank you for your consideration. The error has been corrected.
- Some graphs such as Figures 4 and 5, show error bars that need to mention the reputation number or change the graph to the “scatter plot with error bar”.
In this case, we did our best to match comment and manuscript but unfortunately we do not fully understand what we have to do in this comment. In the mentioned Figures, error bars are shown but we do not know what reputation number means in terms of error bars. Can the reviewer clarify this sentence for us? Thank you.
- It is recommended to do a comparison study or provide a table and compare the recent azo dye removal capacity of adsorbent with your work to emphasize the advantages of your work. You can use similar studies like the following refs. 10.1016/j.jhazmat.2019.01.107 and 10.1016/j.molliq.2018.12.050
Thank you for your comment. A comparison between different polymers can be observed in Table 3, we have just compared the goodness of our adsorbents towards the same azo dye (Direct Red), for this reason a new comparison was not conceived in this regard. This increases the length and the number of new references, in fact we have more than 50 references in the new version of the manuscript.
- Synthesis process, adsorption, and further photodegradation can be illustrated as a graphical abstract to increase the impact of the work. Same references as comment number 10 can help you to come up with a good idea.
Thank you for your comment. A new graphic abstrac has been developed.
Pay attention to connectors and revised the whole manuscript to improve its English of it. And correct the typos. For example “synthetized” in line 38 should change to “synthesized”.
Thank you for your consideration. The English of the article has been reviewed by an expert.

Round 2
Reviewer 1 Report
I have reviewed the corrected version of the manuscript, and it has improved significantly. The manuscript can be accepted for publication with the following modifications.
I suggest unifying the format of all the graphics.
Figure 5 is not readable. If you cannot improve it, you can send it to the support information section.
It is still necessary to update the bibliography regarding cyclodextrin polymers with magnetic nanoparticles.
Author Response
I have reviewed the corrected version of the manuscript, and it has improved significantly. The manuscript can be accepted for publication with the following modifications.
Dear reviewer, thank you for your comments. We followed your advice to improve the present manuscript.
- I suggest unifying the format of all the graphics.
Thank you for your comment. We have analysed our graphs and we have tried to unify the overall format of them according to the type of results presented in them.
- Figure 5 is not readable. If you cannot improve it, you can send it to the support information section.
We do not know what happened with Figure 5. In the docx document the figure can be observed without a problem, however when we analysed the PDF file we noticed that lines were missing and for sure the figure cannot be interpreted under these conditions. We are going to do our best to include Figure 5 correctly.
- It is still necessary to update the bibliography regarding cyclodextrin polymers with magnetic nanoparticles.
Thank you for your recommendation. Following your suggestion we have conducted a thorough search for cyclodextrin polymers coated with magnetic nanoparticles. The following articles have been included in the new version of our manuscript. The whole reference list has been updated accordingly.
Moradi Shahrebabak, S., Saber-Tehrani, M., Faraji, M., Shabanian, M., & Aberoomand-Azar, P. (2020). Magnetic solid phase extraction based on poly (β-cyclodextrin-ester) functionalized silica-coated magnetic nanoparticles (NPs) for simultaneous extraction of the malachite green and crystal violet from aqueous samples. Environmental monitoring and assessment, 192(5), 1-13. https://doi.org/10.1007/s10661-020-8185-6
Xu, J., Tian, Y., Li, Z., Tan, B. H., Tang, K. Y., & Tam, K. C. (2022). β-Cyclodextrin functionalized magnetic nanoparticles for the removal of pharmaceutical residues in drinking water. Journal of Industrial and Engineering Chemistry, 109, 461-474. https://doi.org/10.1016/j.jiec.2022.02.032
Juengchareonpoon, K., Wanichpongpan, P. & Boonamnuayvitaya, V. Functionalization of magnetite nanoparticles with carboxymethyl-beta-cyclodextrin for oxytetracycline removal. Appl. Phys. A 127, 197 (2021). https://doi.org/10.1007/s00339-021-04320-3.
Hassan, M., Naidu, R., Du, J., Qi, F., Ahsan, M. A., & Liu, Y. (2022). Magnetic responsive mesoporous alginate/β-cyclodextrin polymer beads enhance selectivity and adsorption of heavy metal ions. International Journal of Biological Macromolecules, 207, 826-840. https://doi.org/10.1016/j.ijbiomac.2022.03.159.
Chen, D., Shen, Y., Wang, S., Chen, X., Cao, X., Wang, Z., & Li, Y. (2021). Efficient removal of various coexisting organic pollutants in water based on β-cyclodextrin polymer modified flower-like Fe3O4 particles. Journal of Colloid and Interface Science, 589, 217-228. https://doi.org/10.1016/j.jcis.2020.12.109.
Zhang, S., Ange, K. U., Ali, N., Yang, Y., Khan, A., Ali, F., ... & Bilal, M. (2022). Analytical perspective and environmental remediation potentials of magnetic composite nanosorbents―A review. Chemosphere, 135312. https://doi.org/10.1016/j.chemosphere.2022.135312
Bayatloo, M. R., & Nojavan, S. (2022). Rapid and simple magnetic solid-phase extraction of bisphenol A from bottled water, baby bottle, and urine samples using green magnetic hydroxyapatite/β-cyclodextrin polymer nanocomposite. Microchemical Journal, 175, 107180. https://doi.org/10.1016/j.microc.2022.107180
Moradi, O., & Sharma, G. (2021). Emerging novel polymeric adsorbents for removing dyes from wastewater: A comprehensive review and comparison with other adsorbents. Environmental Research, 201, 111534. https://doi.org/10.1016/j.envres.2021.111534
Nkinahamira, F., Alsbaiee, A., Zeng, Q., Li, Y., Zhang, Y., Feng, M., ... & Sun, Q. (2020). Selective and fast recovery of rare earth elements from industrial wastewater by porous β-cyclodextrin and magnetic β-cyclodextrin polymers. Water Research, 181, 115857. https://doi.org/10.1016/j.watres.2020.115857
Salazar, S., Yutronic, N., & Jara, P. (2020). Magnetic β-Cyclodextrin nanosponges for potential application in the removal of the neonicotinoid dinotefuran from wastewater. International Journal of Molecular Sciences, 21(11), 4079. https://doi.org/10.3390/ijms21114079.

Reviewer 3 Report
1- The Y axis of Figure 4 is not written.
2- The style and Font size of axis titles for each figure are different. For example, figures 4 and 5 ad 6.
3- Thew authors should revise the equation orders. Equation number r10 is in line 413 and the authors used number 1 for the next equation.
4- A very high portion of the citations are from more than 10 and 15 years ago (even some references are for the 90s). It is highly recommended to use the most recent references related to your work (last 5 years).
Author Response
Dear reviewer, thank you for your comments. We followed your advice to improve the present manuscript.
- The Y axis of Figure 4 is not written.
Thank you for your comment. The figure has been improved and the format unified with respect to the others.
- The style and Font size of axis titles for each figure are different. For example, figures 4 and 5 ad 6.
Thank you for your comment. We have analysed our graphs and we have tried to unify the overall format of them according to the type of results presented in them.
- Thew authors should revise the equation orders. Equation number r10 is in line 413 and the authors used number 1 for the next equation.
Thank you for your recommendation. We have revised the document, and organized the numbering of the equations.
- A very high portion of the citations are from more than 10 and 15 years ago (even some references are for the 90s). It is highly recommended to use the most recent references related to your work (last 5 years).
Thank you for your comment. It is true that some references are old, but they are the ones that refer to kinetic analysis. Relevant references have been included in the revised version of the manuscript. These new references are from 2020 to 2022, below are the new references added to the introduction and discussion section.
Moradi Shahrebabak, S., Saber-Tehrani, M., Faraji, M., Shabanian, M., & Aberoomand-Azar, P. (2020). Magnetic solid phase extraction based on poly (β-cyclodextrin-ester) functionalized silica-coated magnetic nanoparticles (NPs) for simultaneous extraction of the malachite green and crystal violet from aqueous samples. Environmental monitoring and assessment, 192(5), 1-13. https://doi.org/10.1007/s10661-020-8185-6
Xu, J., Tian, Y., Li, Z., Tan, B. H., Tang, K. Y., & Tam, K. C. (2022). β-Cyclodextrin functionalized magnetic nanoparticles for the removal of pharmaceutical residues in drinking water. Journal of Industrial and Engineering Chemistry, 109, 461-474. https://doi.org/10.1016/j.jiec.2022.02.032
Juengchareonpoon, K., Wanichpongpan, P. & Boonamnuayvitaya, V. Functionalization of magnetite nanoparticles with carboxymethyl-beta-cyclodextrin for oxytetracycline removal. Appl. Phys. A 127, 197 (2021). https://doi.org/10.1007/s00339-021-04320-3.
Hassan, M., Naidu, R., Du, J., Qi, F., Ahsan, M. A., & Liu, Y. (2022). Magnetic responsive mesoporous alginate/β-cyclodextrin polymer beads enhance selectivity and adsorption of heavy metal ions. International Journal of Biological Macromolecules, 207, 826-840. https://doi.org/10.1016/j.ijbiomac.2022.03.159.
Chen, D., Shen, Y., Wang, S., Chen, X., Cao, X., Wang, Z., & Li, Y. (2021). Efficient removal of various coexisting organic pollutants in water based on β-cyclodextrin polymer modified flower-like Fe3O4 particles. Journal of Colloid and Interface Science, 589, 217-228. https://doi.org/10.1016/j.jcis.2020.12.109.
Zhang, S., Ange, K. U., Ali, N., Yang, Y., Khan, A., Ali, F., ... & Bilal, M. (2022). Analytical perspective and environmental remediation potentials of magnetic composite nanosorbents―A review. Chemosphere, 135312. https://doi.org/10.1016/j.chemosphere.2022.135312
Bayatloo, M. R., & Nojavan, S. (2022). Rapid and simple magnetic solid-phase extraction of bisphenol A from bottled water, baby bottle, and urine samples using green magnetic hydroxyapatite/β-cyclodextrin polymer nanocomposite. Microchemical Journal, 175, 107180. https://doi.org/10.1016/j.microc.2022.107180
Moradi, O., & Sharma, G. (2021). Emerging novel polymeric adsorbents for removing dyes from wastewater: A comprehensive review and comparison with other adsorbents. Environmental Research, 201, 111534. https://doi.org/10.1016/j.envres.2021.111534
Nkinahamira, F., Alsbaiee, A., Zeng, Q., Li, Y., Zhang, Y., Feng, M., ... & Sun, Q. (2020). Selective and fast recovery of rare earth elements from industrial wastewater by porous β-cyclodextrin and magnetic β-cyclodextrin polymers. Water Research, 181, 115857. https://doi.org/10.1016/j.watres.2020.115857
Salazar, S., Yutronic, N., & Jara, P. (2020). Magnetic β-Cyclodextrin nanosponges for potential application in the removal of the neonicotinoid dinotefuran from wastewater. International Journal of Molecular Sciences, 21(11), 4079. https://doi.org/10.3390/ijms21114079.
